# Optimizing Operation Room Utilization—A Prediction Model

**Benyamine Abbou** [1,2,*,†], **Orna Tal** [3,4,5,†], **Gil Frenkel** [6], **Robyn Rubin** [6] **and Nadav Rappoport** [6,7,*]

1    Medical Management Department, Hillel-Yaffe Medical Center, Hadera 3810101, Israel
2    Ruth & Bruce Rappaport Faculty of Medicine, Technion—Israel Institute of Technology, Haifa 3200003, Israel
3    Medical Management Department, Shamir Medical Center (Assaf Harofeh), Be'er Ya'akov 7033001, Israel; ornatal@shamir.gov.il
4    Faculty of Management, Bar Ilan University, Ramat Gan 5290002, Israel
5    The Israeli Center for Emerging Technologies (ICET), Be'er Ya'akov 7033001, Israel
6    Division of Government Medical Centers, Israeli Ministry of Health, Jerusalem 9101002, Israel; gilor9101@gmail.com (G.F.); robyn.rubin@moh.gov.il (R.R.)
7    Department of Software and Information Systems Engineering, Ben-Gurion University of the Negev, Be'er Sheva 8410501, Israel
*    Correspondence: benyaminea@hymc.gov.il (B.A.); nadavrap@bgu.ac.il (N.R.); Tel.: +972-4774-4228 (B.A.); +972-74779-5156 (N.R.)
†    These authors contributed equally to this work.

**Abstract:** Background: Operating rooms are the core of hospitals. They are a primary source of revenue and are often seen as one of the bottlenecks in the medical system. Many efforts are made to increase throughput, reduce costs, and maximize incomes, as well as optimize clinical outcomes and patient satisfaction. We trained a predictive model on the length of surgeries to improve the productivity and utility of operative rooms in general hospitals. Methods: We collected clinical and administrative data for the last 10 years from two large general public hospitals in Israel. We trained a machine learning model to give the expected length of surgery using pre-operative data. These data included diagnoses, laboratory tests, risk factors, demographics, procedures, anesthesia type, and the main surgeon's level of experience. We compared our model to a naïve model that represented current practice. Findings: Our prediction model achieved better performance than the naïve model and explained almost 70% of the variance in surgery durations. Interpretation: A machine learning-based model can be a useful approach for increasing operating room utilization. Among the most important factors were the type of procedures and the main surgeon's level of experience. The model enables the harmonizing of hospital productivity through wise scheduling and matching suitable teams for a variety of clinical procedures for the benefit of the individual patient and the system as a whole.

**Keywords:** surgery; electronic health records (EHR); prediction model; operation room (OR); machine learning



## 1. Introduction

Conducting general surgical activity within the framework of a public general hospital is a challenge; it involves a highly complex ensemble composed of multidisciplinary teams, working in a risky, overloaded, uncertain environment. Yet, it encompasses promises for better health solutions for the patient and professional quality branding for the hospital in a competitive health arena. Moreover, operating rooms are the most important source of both income and expense for hospitals and the most profitable path for the supplier. Therefore, hospital managers are motivated to increase the effectiveness of schedules and plans [1]. The productivity of operating rooms influences not only the performance-related financial status of hospitals [2] but may also affect service quality and patient satisfaction.

Many elements may affect the effective synchronization of hospital surgical activity, especially when multiple surgical theaters (STs) are involved. These include operational

space, equipment, scheduling, various combinations of skilled teams, personal contracts with workers, and costs. In a public healthcare system with complete insurance coverage such as in Israel, no selection of cases occurs, so the case mix is unpredictable. Hence, surgical activity is mostly dictated by clinical urgency and medical guidelines.

Factors that influence operating room (OR) performance include the characteristic of the surgery (elective/non-elective) and performance criteria such as waiting time (patient/surgeon), utilization and flow (OR, ICU), overtime, completion time and patient postponement/rejection and preferences, as well as humanitarian-ethical aspects and financial assets [1].

Above all, there is the need to deal with unexpected changes in the operational program as well as urgent patients who burst into the scheduled program, on top of cancellations due to either a patient's condition or hospital constraints. Thus, ST administrators tend to limit activity to the most minimal timeframe possible, yielding insufficient utilization and productivity.

A relatively simple method to organize OR utilization is to plan a daily or weekly program by the cumulative length of sequencing procedures. The duration of each procedure may be influenced by factors related to the patient or their caregivers, and daily planning should integrate all these factors, as well as the average or median time for each of the procedures. A successful plan is often a reflection of an OR administrator's long-term experience.

Currently, a sufficient OR utilization plan is achieved about one-third of the time. The rest are shorter or longer than optimal utilization and far from maximizing hospital capacity [3]. These gaps not only harm patients by delaying treatment but also reduce income [4] and patient satisfaction [5], as well as overloading hospital personnel. Even in a publicly funded health system, managers cannot ignore the potential benefit embedded in increasing surgical activity, to enhance income and branding.

In the United States, nearly a third of all surgeries are overlapping surgeries, which are defined as two surgeries performed by the same surgeon that overlap in time. This policy has received significant legal and public scrutiny in aspects related to safety and transparency [6], and therefore some hospitals have revised administrative policies, eliminating the prospect of overlapping surgeries [7]. The change in policy on the subject has led to a decrease in the efficiency and outputs of hospitals and has far-reaching economic consequences [8]. It is therefore a prerequisite to achieve a more accurate OR schedule for predicting the preliminary times of each individual surgery that can serve as a significant tool for improving efficiency and productivity in the new legal situation.

OR scheduling is traditionally based on a surgeon's self-estimation of operation duration, yet the accuracy of these evaluations is far from being sufficient to enhance performance [9]. Solutions to maximize OR performance have been discussed. Di Martinelly suggested a model focusing on the relationship between the operating room and nurse management, while considering resource constraints [10]. Various mathematical models have been offered: Hanset proposed a theoretical surgical journey simulation that involves estimating an optimal time slot for each operation, focusing on technical solutions such as operation starting time and recovery bed availability [11]. Pham suggested using multimode job shop or blocking [12]. Lin developed an artificial bee colony algorithm to quickly find feasible solutions based on the earliest due date and longest processing time rules [13]. Technical, managerial tools have also been reported, such as a dashboard to assess workflows of patients to enhance evaluation and prioritization, using hierarchic filters of date, time, duration, location, surgeon characteristics, and procedure features [14]. A Monte Carlo simulation was used to guide decisions on balancing resources for elective and non-elective surgical procedures [15]. And finally, Bartek et al. developed predictive models based on average historic procedure time and surgeon estimates [16]. They used single-site data to develop a model per surgeon and per service, which limits the usage of such a model for only the 12 services and 93 surgeons used.

Different ML-based approaches have been developed to pursue an advanced methodology for managing OR utilization. ML is the field of computer algorithms that can perform a task based on experience [17]. The advantage of ML-based models vs. rule-based models is that ML models are data-driven and not knowledge-driven. In addition, ML-based models can take into account complex relationships between data points. The computational models are trained on large datasets representing past experience and are driven by algorithms to accurately predict unknown labels or future events. An example of such prediction models is diagnosing disease from X-ray images [18]. Furthermore, it has been found that implementing ML algorithms that detect intracranial hemorrhage on non-contrast-enhanced head CT studies within the clinical workflow reduced wait time, and thus overall turnaround time, when specifically used to prioritize examinations [19].

However, while adopting artificial intelligence (AI) in imaging focuses on analyzing a constellation of unstructured data (volume), implementing AI in operating room planning strategy differs slightly from our model, which is based on multi-dimensional data that change over time (variety, velocity, and volume). Recently, ML models for OR facilities were developed: for example, to analyze the key factors that affect the identification of surgeries with high cancellation risk [20]; to improve the accuracy of duration prediction of complex surgical activities, such as the duration of robot-assisted surgery [21]; and to improve the whole surgical workflow [22]. ML can also be used for improving surgery training [23]. Our assumption is that OR performances may be measured, analyzed, and assessed through a wise model to achieve better consequences, that is, to enable health managers to identify pathways to improve patient outcomes, increase the quality of care, and even prioritize OR activities through the lens of economic benefit as well.

In order to increase OR utilization, we developed an ML model for predicting surgery duration using pre-operative data of the coming operations. The scheduling for elective surgeries was done on the day before surgery. We collected relevant data from two general public hospitals in Israel, trained the models, and evaluated performances.

## 2. Methods

### 2.1. Data Source

The data are an extract of the electronic health records (EHR) from two general hospitals: Hillel-Yaffe (HY) and Shamir (SH). They are both public hospitals owned and managed by the Israeli Ministry of Health, with 515 and 891 beds, respectively. Both HY and SH are single-site medical centers. Both treat the general population in their region without restriction to specific health care insurance providers, meaning no cream skimming or population bias exists. The data contain all surgeries that occurred from December 2009 to May 2020 in these two medical centers.

### 2.2. Outcome Measures and Predictors

Surgery length was defined as the difference between the time when the patient entered the operating room and when they left it. An alternative definition for the duration of the operation is the time from the first cut to closure, but this was not used here, as we also wanted to estimate the time of logistical preparation in the operation room, including anesthetic procedures, thereby estimating the actual OR occupancy slot rather than surgeon–procedure performance time.

### 2.3. Data Cleaning and Preprocessing

The features for the prediction model were clinical data from the EHR, and we used only the data that was available before the surgery took place. Our dataset covered demographic data such as age, gender, country of origin, marital status, number of children, religion, and weight. It also included operative plan data such as procedures, localization, type of anesthesia, number of surgeons, and the main surgeon's level of experience (hours in the operating room and number of operations participated in). In addition, it included clinical data such as previous diagnoses, drug prescriptions, last laboratory test results,

smoking status, and risk factors (Supplementary Table S1). The source data included 13,520 types of diagnoses in ICD9-CM codes. We grouped them into 282 higher-order single-level disease categories using Clinical Classifications Software (CCS) (Agency for Healthcare Research and Quality (AHRQ), Rockville, MD, USA) [24]. The number of different drugs administrated was 3698. Drugs were grouped into 930 categories based on matching Anatomical Therapeutic Chemical (ATC) 5th level. The complete list of features is given in Supplementary Table S1.

Overall, 122,439 surgeries data were extracted from HY and 175,041 from SH EHRs. We excluded samples with missing data regarding the operating room, time of entrance or time of exit from the OR (1491 surgeries), gender, age, or main surgeon (205). We also excluded surgeries with possible bias of an atypical manner, such as surgeries performed outside the planning schedule (i.e., from 7 PM to 7 AM) (9529 HY, 12,929 SH) or performed during the weekend (Friday or Saturday) (7778 HY, 10,346 SH), as well as surgeries shorter than 10 min (1874 HY, 1719 SH). Overall, 102,301 (149,308) surgeries of 77,643 (119,525) unique patients from HY (SH) were used in our analysis (Figure 1).

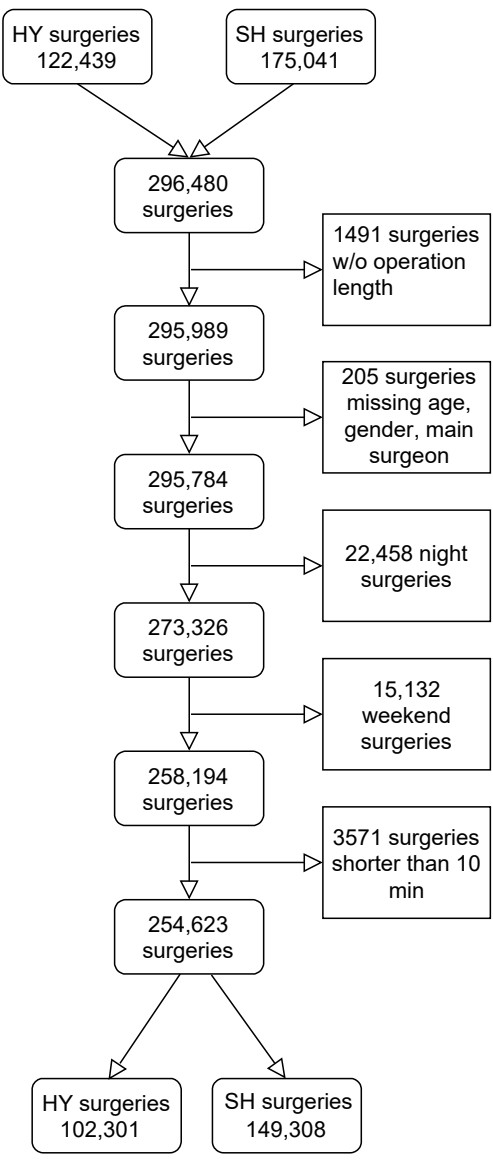

**Figure 1.** Data selection flow chart.

Missing data were not imputed, as the two types of prediction models we used handled missing data. The naïve model (described below) predicts the median length for the procedures and therefore was not affected by missing values. The XGBoost model is based on decision trees that handle missing values by adding branches for such values [25,26].

Continuous variables were standardized by reducing the mean value and dividing by the standard deviation of the training samples. Categorical variables such as previous diagnoses and procedures were represented using a one-hot encoding, such that every diagnosis or procedure was represented as a binary feature that indicated whether a specific diagnosis or procedure was recorded in this sample or not.

### 2.4. Models' Training

Because data sharing was impracticable due to indecisive regulation, we repeated the analysis, including data extraction, cleaning, training, and testing the models independently, in each medical center. We split the data to train and test based on the year of operation. Such a split procedure was intended to mimic real-world scenarios, where models are trained on past data and evaluated on surgeries occurring afterward. Surgeries performed before 2018 were used to train the model, and the others were used as the testing set. We trained two types of models. As a baseline, we trained a naïve model based on the median length of similar surgeries. For a given surgery with a specific set of procedures, the naïve model predicted the duration of this surgery to be the median duration of all surgeries in the training data having the same set of procedures. The naïve model did not consider other factors and parameters of the surgery or the patient. Therefore, missing factors did not affect the prediction.

The second type of model was an eXtreme Gradient Boosting (XGBoost) model based on the gradient boosting framework of multiple trees [25]. We used the training data to optimize the hyperparameters of the XGBoost model using Bayesian search and 5-fold cross-validation. Bayesian optimization uses a posterior distribution of optimization function, and as the number of hyperparameters it tries increases, the distribution function becomes more competent at finding the optimal hyperparameters. We used the BayesianOptimization function in the bayes_opt Python package [27] for hyperparameter optimization in our Bayesian approach [28]. The list of parameters that were optimized, their range, and the optimal values are described in Table 1.

**Table 1.** XGBoost Hyperparameter tuning. The list of parameters, the description, and the name of the parameter in the XGBoost package are given in the table in addition to the range in which the parameters were optimized as well as the optimal values in SH and HY.

| Parameter Type | Parameter Name in XGBoost Package | Range of Search | Optimal Value in SH | Optimal Value in HY |
|---|---|---|---|---|
| Subsample ratio of columns when constructing each tree | colsample_bytree | 0.6–1 | 0.713555 | 0.991201 |
| Minimum loss reduction required to make a further partition on a leaf node of the tree | gamma | 0–5 | 2.206600 | 1.073363 |
| Step size weight shrinkage | learning_rate | 0.01–1 | 0.247214 | 0.271243 |
| Maximum depth of a tree | max_depth | 3–6 | 5 | 5 |
| Minimum sum of instance weight needed in a child | min_child_weight | 1–10 | 5.427004 | 1.240320 |
| Number of trees | n_estimators | 100–1000 | 762 | 486 |
| Subsample ratio of instances | subsample | 0.6–1 | 0.767184 | 0.818254 |

### 2.5. Evaluation Metrics

Surgery lengths vary for different procedures, and every evaluation metric has limitations. Therefore, we used several evaluation metrics.

The root mean squared error (RMSE) is the square root of the average of squared errors:

$$RMSE(\hat{y}, y) = \sqrt{\frac{\sum_{i=1}^{n}(\hat{y}_i - y_i)^2}{n}}$$

where $\hat{y}$ is the predictions vector, $y$ is the vector with true labels, and $n$ is the number of samples.

The disadvantage here is that RMSE gives higher weights to larger errors, as the error is squared.

The mean absolute error (MAE) is the average of absolute errors.

$$MAE(\hat{y}, y) = \frac{1}{n}\sum_{i=1}^{n}|\hat{y}_i - y_i|$$

The explained variance (EV) is the fraction of the model's total variance explained by the present factors. In regressions, it is also called the coefficient of determination ($R^2$).

The mean absolute percentage error (MAPE) is the average of normalized errors where each prediction's error divided by the actual label.

$$MAPE(\hat{y}, y) = \frac{1}{n}\sum_{i=1}^{n}\left|\frac{\hat{y}_i - y_i}{y_i}\right|$$

The mean $\log_2$ ratio (ML2R) is the mean of log in base 2 of the ratio of observed vs. expected length of surgery.

$$ML2R(\hat{y}, y) = \frac{1}{n}\sum_{i=1}^{n}\log_2\left|\frac{y_i}{\hat{y}_i}\right|$$

## 3. Results

### 3.1. Data Sets

In total, the data of 121,539 and 174,450 surgeries were extracted from HY and SH from 77,643 and 119,525 patients, respectively (Figure 1). Slightly more than half of the surgeries were performed on females (59% and 51% in HY and SH). The number of unique procedure types was 3544 in HY and 4721 in SH. Data were extracted from December 2009 to May 2020. The number of surgical departments was 17 in HY and 30 in SH, and the number of surgeons was 580 in HY and 983 in SH. The average surgeon's age was 46.2 in HY and 49.5 in SH. The average number of diagnoses per patient prior to surgery was 8.62 in HY and 8.86 in SH. The average surgery duration was 67.85 min in HY and 81.73 min in SH (Table 2 & Figure 2). The range of surgery durations in HY was 10 to 939 min (median 52.45 and average of 67.85 min) and 10 to 1184 min (median 60.95 and average of 81.73 min) in SH.

Using a density plot of surgery length, we can see that the distribution has a long right tail, with very few surgeries that took a very long time. A similar pattern is seen in both hospitals, yet the average surgical length in HY was shorter (Figure 2).

Surgery length distribution varied across the surgical units and medical centers (Figure 3). Slight differences in the number of surgical departments (20 in HY, 26 in SH) were found. Accordingly, the intensity of performance based on the number and seniority of acting physicians is presented.

**Table 2.** Statistical summary of the data used in the study, stratified by hospital. N: Number of surgeries. IQR: Interquartile Range.

|  | HY | SH |
|---|---|---|
| N | 121,539 | 174,450 |
| **Demographic** | | |
| Age (median, IQR) | 44 (30–64) | 51 (29–68) |
| Females (%) | 59.4 | 50.0 |
| **Preoperative** | | |
| Number of drugs (median, IQR) | 9 (4–17) | 9 (4–18) |
| Number of diagnoses (median, IQR) | 6 (3–11) | 6 (3–11) |
| **Surgeon's experience** | | |
| Number of previous surgeries (median, IQR) | 432 (158–863) | 361 (133–775) |
| Total hours in operating room (median, IQR) | 435.55 (154–892) | 428 (155–963) |
| **Surgery** | | |
| Number of procedures (median, IQR) | 1 (1–1) | 1 (1–1) |
| Operating time in minutes (median, IQR) | 52.45 (31–85) | 60.95 (38–102) |

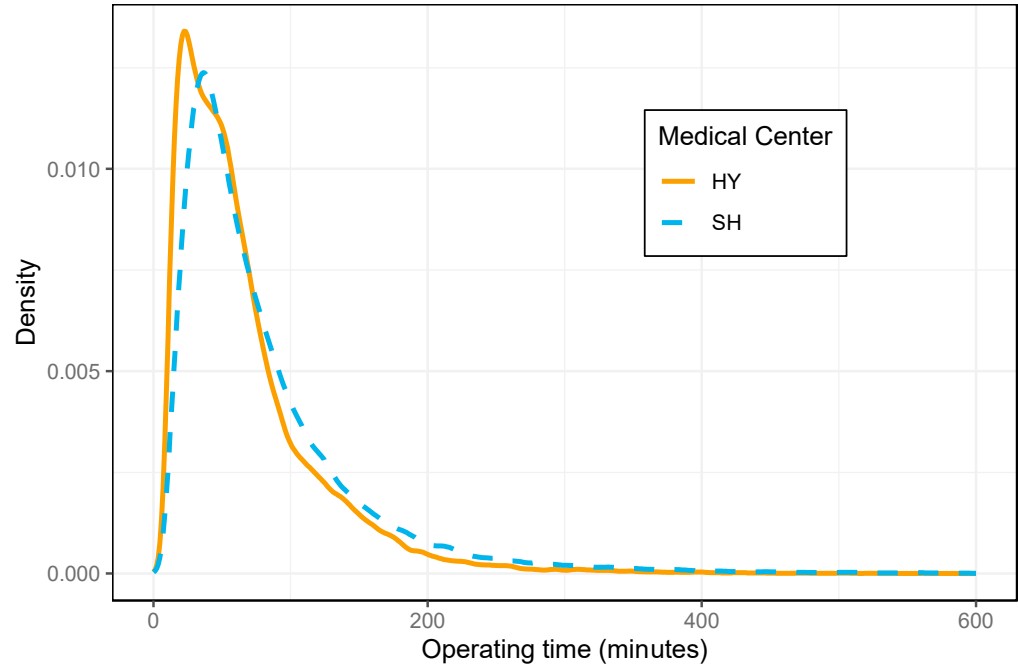

**Figure 2.** Density plot of surgery length per medical center.

*3.2. Model Development*

Data was split to train and test such that surgeries from 2018 and on were only used for testing. In SH, 113,594 (77.04%) surgeries were used for training and 35,714 surgeries (22.96%) for testing. In HY, 91,240 (76.67%) surgeries were used for training and 27,756 (23.33%) for testing.

We evaluated the feature importance of the model based on the F-score [29], which is a common way of estimating a feature's importance. In short, it is the number of times a feature was used to split data in all trees. Feature importance was computed separately for each hospital, meaning that we trained, tested, and evaluated one model for HY and one for SH. The top six most important features in the two models (for both HY and SH, in the same order of importance) were: the main surgeon's experience (in number of surgeries previously conducted), the patient's age, the number of surgeons assigned to the surgery, the number of diagnoses, the number of drugs, and the number of planned procedures.

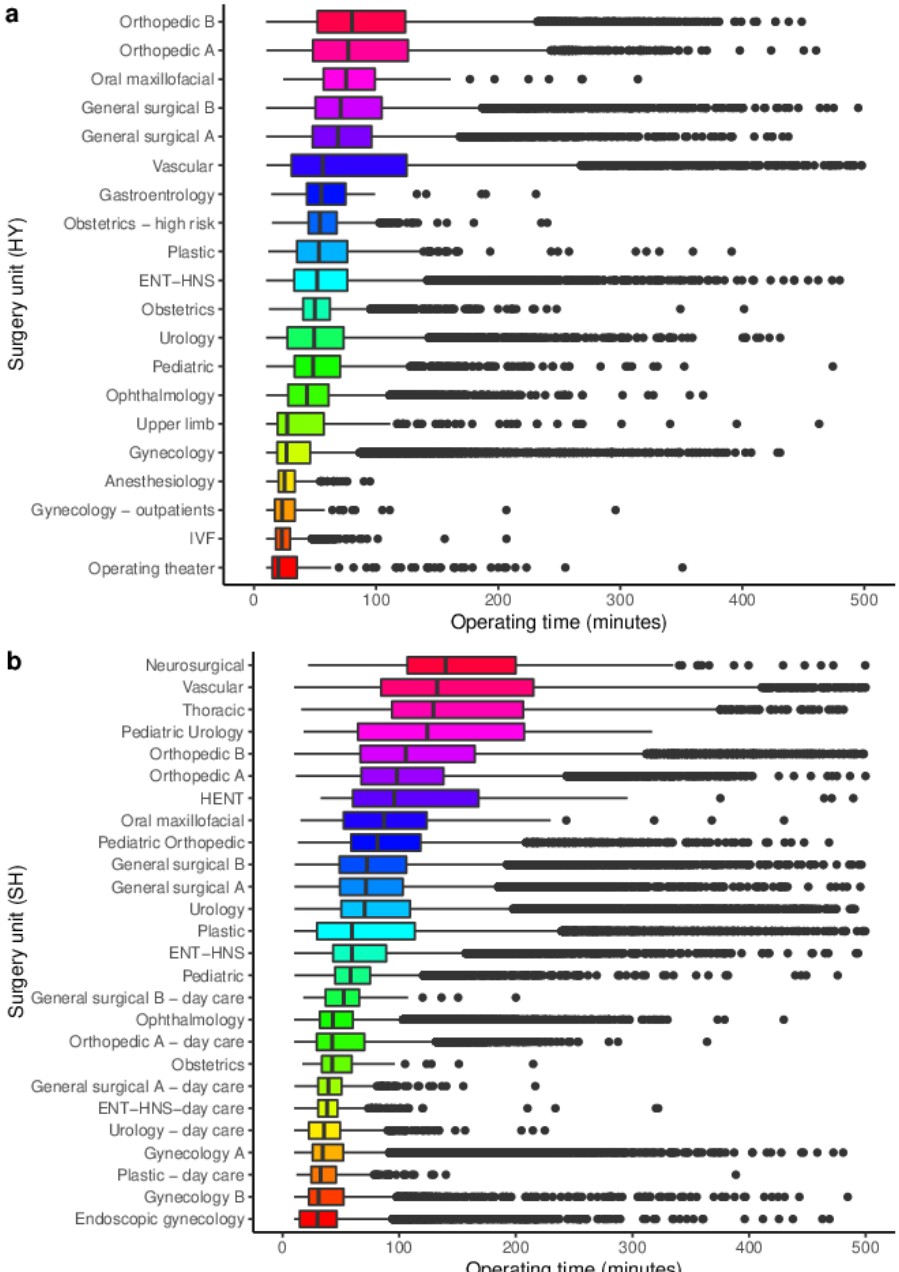

**Figure 3.** Box plots representing the distributions of surgery lengths by surgical units and by hospital, (**a**) HY and (**b**) SH. Each box represents the 3rd to 4th quartile range, and the vertical bar represents the median duration. Surgeries longer than 500 min were removed from this figure. Surgical units were sorted on the Y-axis by median duration.

*3.3. Model Performance*

The performances of the models are summarized in Table 3. Using different measures of performance evaluations, the XGBoost models performed better than the naïve models: the MAE was 21.5 compared to 25.4 in HY and 25.3 compared to 28.7 in SH; RMSE, 36.6 vs. 49.0 (HY), 40.3 vs. 55.0 (SH); PVE, 66.7 vs. 44.0 (HY), 70. vs. 46.8 (SH); and ML2R, 0.46 vs. 0.53 (HY) and 0.46 vs. 0.49 (SH). In the case of MAPE, differences between the naïve and the ML-based model were minor—35.15 vs. 35.37 in HY and 35.09 vs. 32.48 in SH according to hospital performance evaluations.

**Table 3.** Models' performances on the test set.

| Hospital | HY | | SH | |
|---|---|---|---|---|
| N | 27,752 | | 39,468 | |
| Median length | 54.06 | | 67.35 | |
| Model | Naïve | XGB | Naïve | XGB |
| MAE | 25.44 | **21.52** | 28.69 | **25.23** |
| RMSE | 49.03 | **36.64** | 55.03 | **40.26** |
| MAPE | 35.36 | **35.16** | 32.48 | 35.11 |
| PVE | 44.02 | **66.71** | 46.75 | **69.97** |
| ML2R | 0.14 | **−0.05** | 0.14 | **−0.06** |
| AbsErr ≤ 10 min | 40.48 | **40.95** | 36.79 | 32.89 |
| AbsErr ≤ 20 min | 63.18 | **65.56** | 59.76 | 57.25 |
| AbsErr ≤ 10% | 21.03 | **22.49** | 21.93 | 21.69 |
| AbsErr ≤ 20% | 39.63 | **42.65** | 42.41 | 41.21 |

N: number of samples in the test set; MAE: mean absolute error; RMSE: root mean squared error; MAPE: mean absolute percent error; PVE: percent variance explained; ML2R: mean of base 2 log of the ratio of observed and predicted lengths; AbsErr ≤ 10 min: percent of surgeries with predicted error less than or equal to 10 min; AbsErr ≤ 10%: percent of surgeries with absolute error smaller or equal to 10% of observed length. Bold face marks the model with best performance according to each evaluation metric in each medical center.

Due to the variety in typical or average surgery length between surgery units and surgery types, we evaluated the same model stratified by surgery unit and by procedure. Different evaluation measurements were biased by the duration of surgeries. For example, MAE and RMSE had on average higher rates of error for more lengthy procedures. In Figure 4, we plotted the performances according to different measurements by the median length of surgery of that unit. MAPE, PVE, and ML2R were much less affected by the duration of surgeries. As shown in Figure 4, the average MAE and RMSE per surgery unit were highly correlated with the unit's median surgery duration, whereas MAPSE, PVE, and ML2R were poorly correlated with median surgery duration. This was due to the fact the last three included normalization (see Methods).

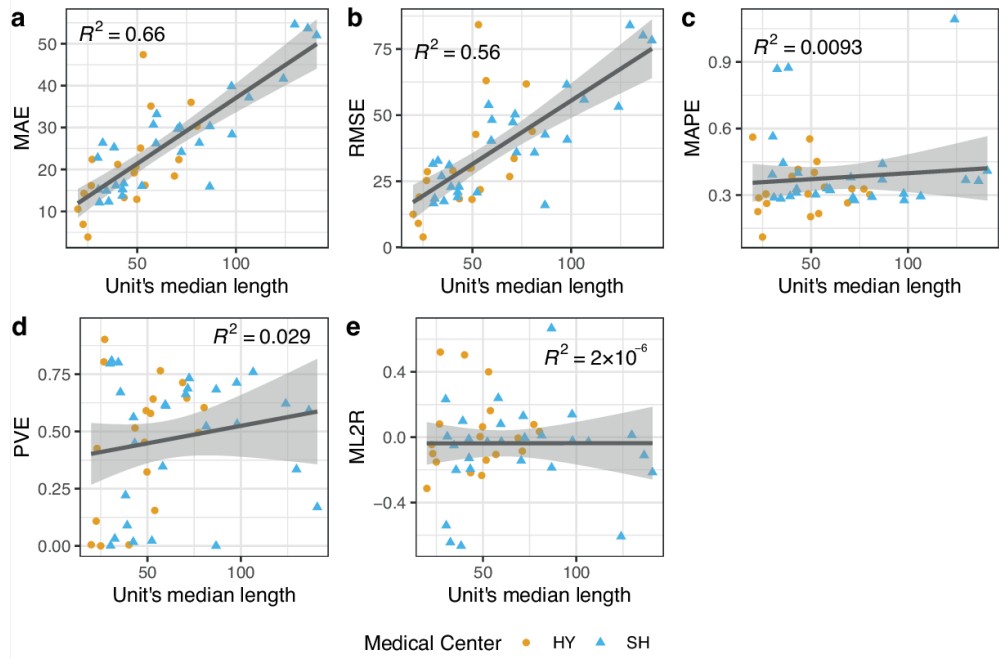

**Figure 4.** Average performance per surgery unit, by median surgery length. For each surgery unit, we computed the median surgery length and the average performance of the model in (**a**) MAE; (**b**) RMSE; (**c**) MAPE; (**d**) PVE, and (**e**) ML2R.

## 4. Discussion

To optimize the utilization of operating rooms, managers and researchers need to balance the productivity and effectiveness of staff alongside the available resources. Other investigators have already developed technical and mathematical models to address this challenge, mainly in a theoretical manner. We present a novel predictive model based on accumulative data from two large medical centers. In contrast to other, previous works [16,30], we used different methods to achieve the same goal of optimizing the prediction of time usage in the operating room. We focused on patients' clinical parameters including diagnoses, laboratory tests, risk factors, and more, as well as surgeons' cumulative experience over 7 years.

The performances of the models are summarized in Table 3. Using different measures of performance evaluation, XGBoost models performed better than the naïve model except for MAPE, in which we found only a slight difference between the naïve and the ML-based model. One drawback of MAPE is that it is not symmetric. For example, if the actual length of a surgery is 10 min and the prediction is 100 min, then the MAPE is $\frac{|10-100|}{10} = 900\%$, while in the symmetric case, where the actual length is 100 min and the prediction is 10 min, the MAPE is $\frac{|100-10|}{100} = 90\%$. Therefore, the same absolute error gives a MAPE of an order of magnitude larger.

The length of surgeries varies across surgery units (Figure 3) and surgery types. Therefore, a prediction that is 10 min off for a surgery that typically takes 20 min is a small error on an absolute scale but a large one on percent scales.

The model is surgeon-based, meaning that the surgeon's experience plays a significant role in prediction, which is in accordance with previous studies [31,32]. Inter-hospital variation among medical personnel (experienced versus inexperienced) will shift the predictive performance curve. Thus, the model may serve as a powerful tool for hospital managers, especially when considering relatively small or rural hospitals. We suggest that this tool enables policymakers to plan strategies that reduce geographic and socio-economic gaps among subpopulations by targeting nationwide hospital human resources. In addition, it can improve human resource allocation and utilization by automating the surgery scheduling that is often done by humans.

Moreover, maximizing OR utilization and minimizing overflow can, in the long run, reduce the load on personnel, improve staff satisfaction, and reduce burnout, and thus expand the benefit to the entire healthcare system beyond enhancing performance. OR managers may consider the benefit of AI as a decision support tool, using simulation-based training assistance [23]. Moreover, hospitals may expect to significantly reduce financial losses with the introduction of policies regulating OR scheduling. This should be conducted wisely to maximize efficiency while still fulfilling the ethical duty to patients [8].

The strengths of our model are that it is based on "real world" accumulative data from two large general hospitals and we used a train and test phase for validation. Our model was trained on a large variety of procedures and no specific surgery types were preselected, other than the exclusion of extremely non-representative surgeries (shorter than 10 min) and those missing data. Moreover, our model is not limited to previously seen procedures or surgeons and can handle missing data. The prediction accuracy of our model was solid, confirming that OR operational performance can be increased by managerial tools. One must bear in mind that we excluded samples with missing data, surgeries performed outside the planning schedule (during a night shift or during the weekend), and surgeries shorter than 10 min, as their predictive value was questionable.

The main limitation of our models is that the model was validated by analyzing only two medical centers. However, we assume that it can be expanded to a national level for decision-makers. The next step is to analyze data from the 11 general public hospitals that are the core network of care providers, thus enabling validation of this model on a national level.

The model's performance varied across surgery units. For example, the unit with more than 50 test samples with the lowest RMSE was found to be the IVF department,

with a RMSE of 6.9 min (Supplementary Table S2). This is probably because the range of different procedures performed there is small, and do not tend to lead to complications. Based on other evaluation metrics, different models achieved the best performance for different departments. For example, the obstetrics department had the highest MAPE, while the pediatrics department had the highest ML2R.

## 5. Conclusions

Surgery length prediction is possible via integrating clinical data and surgeons' level of experience. We anticipate that such a prediction model can improve the utilization of OR resources. Such a model may be more suitable for some surgery units or types of surgeries than others.

In this study we demonstrated the principle wherein big data can be used to better predict the duration of surgery in a general hospital. This study should be seen as a proof of concept. Yet, our model's performance was not optimal for all surgery types and surgical departments. Moreover, the prediction model can be further expanded to other surgical outcomes, such as predicting the length of post-surgery hospitalization, significant complications, and even the success or failure of surgery. These advanced capabilities will have a significant impact worldwide, both on clinical aspects of quality and safety as well as economic aspects. Since our model is surgeon-dependent, it may raise questions regarding personal and professional abilities that may require input by surgeons' professional guilds.

**Supplementary Materials:** The following supporting information can be downloaded at https://www.mdpi.com/article/10.3390/bdcc6030076/s1: Supplementary Table S1: Final list of features used for training the model, Supplementary Table S2: Model performance by surgery unit.

**Author Contributions:** Conceptualization, B.A. and N.R.; methodology, B.A., O.T. and N.R.; software, G.F. and N.R.; validation, O.T., R.R. and N.R.; formal analysis, G.F., O.T. and N.R.; investigation, B.A. and N.R.; resources, B.A. and O.T.; data curation, R.R. and N.R.; writing—original draft preparation, N.R.; writing—review and editing, B.A., O.T., G.F., R.R. and N.R.; visualization, G.F. and N.R.; supervision, B.A. and N.R. All authors have read and agreed to the published version of the manuscript.

**Funding:** This research received no external funding.

**Institutional Review Board Statement:** The study was conducted according to the guidelines of the Declaration of Helsinki and approved by the Institutional Review Boards of Hillel Yafe (protocol code 0093-19-HYMC on 20 August 2019), as well as by the Institutional Review Boards of Shamir Medical Center (protocol code 0308-19-ASF on 3 December 2019).

**Informed Consent Statement:** Patient consent was waived due to the retrospective nature of the study and the data being provided to the researchers in an anonymized format.

**Data Availability Statement:** Data cannot be made available to the public due to its private nature.

**Acknowledgments:** We would like to thank Naama Perry Cohen and Nir Makover from the Division of Government Medical Centers at the Israeli Ministry of Health, whose guidance and support were invaluable for this study to materialize. We would also like to thank Hillel Yaffe and Shamir Medical Center's IT departments and the Kineret team members from the Division of Government Medical Centers at the Israeli Ministry of Health who assisted in data collection and provided other assistance as well.

**Conflicts of Interest:** The authors declare no conflict of interest.

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
