# Peer review of "Optimizing Operation Room Utilization—A Prediction Model"

_2504-2289, doi:10.3390/bdcc6030076_

Round 1

Reviewer 1 Report

This work represents the potential use of ML-based resource management for surgery unit. The study is overly well described. Few questions I have:

1) Table 1 is unclear. Why not considering medium or IQR?

2) what are the reasons for not having the same data splitting between two hospital?

Author Response

Response to Reviewer 1 Comments

We thank the reviewer for his invaluable comments. Please see our point-by-point answer to the comments below.

Point 1: Table 1 is unclear. Why not considering medium or IQR?

Response 1: Indeed, in the original submission we provided the mean and IQR, we changed now Table 1 to report the median and the IQR of the different values.

Point 2: What are the reasons for not having the same data splitting between two hospitals?

Response 2: We have the same split approach for the two hospitals which is to split the samples based on operation year. All surgeries prior to 2018 were used to train the models, and all surgeries operated from 2018 and on were used for testing. Therefore the numbers differ slightly, in SH 77.04% of samples were used to train, while in HY only 76.67%. We clarify that in the text:

Under “Model development” subsection we wrote now “Data was split to train and test such that surgeries from 2018 and on were only used for testing. In SH hospital 113,594 (77.04%) surgeries were used for training and 35,714 surgeries (22.96%) for testing. in HY hospital 91,240 (76.67%) surgeries were used for training and 27,756 (23.33%) for testing.
And under “Models’ training” subsection we wrote now “Surgeries performed before 2018 were used to train the model, and the others were used as the testing set.

Reviewer 2 Report

Authors investigated the question of optimizing operation room utilization, which is critical to patients and hospitals as stated in the manuscript.  Authors collected surgery EHR data from two general hospitals in Israel and compared the performance of the naïve and XGBoost models. 

I read the manuscript with great interest. I have serval comments and hope can improve the paper. 

1.      The modeling activities are based on EHR data from two selected hospitals. This is fine.  But the scope of inference may be limited.  Because the data were not randomly selected from a larger geographical area.   In addition, selection bias may exist.  Why and how did you choose these two?

2.     Line 141, authors mentioned that they excluded samples with missing data.  I would like to see a table describing the summary (e.g. proportion) of missing data.  Missing data is common is medical data analysis, but excluding the missing data directly is a potential source of bias.  

3.     Line 147, ‘The naïve model (described below)’ predicts the median length for the procedures and there is not affected by missing values’’.  Firstly, I do not see authors describe the naïve model explicitly.  What is that?  Please consider providing more details of the naïve model and writing down the equation if possible.  Secondly, this sentence is not true.  There is no causal relationship.  For instance, the simple linear regression for normally distributed outcome will predict the mean of the outcome variable, but missing values will affect the model estimation. 

4.     Line 149, XGBoost model handles the missing values by adding branches for such values.  It is very rare, to me, that treating missing value as a category in the model.  Do you have any reference on this?

5.     Line 156, authors split the data into training and test data by year, which is not randomly splitting. Data in 2018 served as test data.  I do not agree with this approach.  Test data are used to validate the performance of the model.  Training and test data are supposed to be comparable.  By doing your way, the time effect is mixed in the two types of data, which is not expected.

6.      Line 166 to 172, please consider putting them into a table and giving more explanations for these parameters. E.g. what they are, how did you tune them, etc. Otherwise, readers may feel confused.

7.     In Table 1, ‘N’: a comma is missing for SH column. ‘Number of procedures’: what is (1-1)? If it is IQR, then it looks not right. ‘Total time it operating room’: is it ‘in’? what is the unit for this variable?

Thank you.

Author Response

We thank the reviewer for the invaluable comments and suggestions that have led to significant improvement to the manuscript. Please see our point-by-point answer to the comments. We have done our best to address all the comments.

Round 2

Reviewer 2 Report

Thanks for the revision.  I only have few minor comments on Table 2 and Figure 1, 2. Specifically,

(a) It is expected to name the parameters:  i.e., for 'number of drugs', it is better to be 'Mean number of drugs' or 'Number of drugs (mean; SD)' like you did for 'Age'.   Same as the rest of parameters.

(b) 'Females (%)' is a categorical variable, which also has standard deviation (SD). I suggest that add the SD for this variable as well to be consistent with other variables.

(c) Margin of Figure 1 looks not completely displayed.  

(d) Please consider plotting Figure 2 with thicker lines. In R, it is 'lwd' for plot() function.
